# Gravity and Cosmology in Kaniadakis Statistics: Current Status and Future Challenges

**DOI:** 10.3390/e24121712

**Published:** 2022-11-24

**Authors:** Giuseppe Gaetano Luciano

**Affiliations:** Applied Physics Section of Environmental Science Department, Universitat de Lleida, Av. Jaume II, 69, 25001 Lleida, Spain; giuseppegaetano.luciano@udl.cat

**Keywords:** Kaniadakis entropy, relativistic theory, gravity, cosmology, big bang nucleosynthesis

## Abstract

Kaniadakis statistics is a widespread paradigm to describe complex systems in the relativistic realm. Recently, gravitational and cosmological scenarios based on Kaniadakis (κ-deformed) entropy have been considered, leading to generalized models that predict a richer phenomenology comparing to their standard Maxwell–Boltzmann counterparts. The purpose of the present effort is to explore recent advances and future challenges of Gravity and Cosmology in Kaniadakis statistics. More specifically, the first part of the work contains a review of κ-entropy implications on Holographic Dark Energy, Entropic Gravity, Black hole thermodynamics and Loop Quantum Gravity, among others. In the second part, we focus on the study of Big Bang Nucleosynthesis in Kaniadakis Cosmology. By demanding consistency between theoretical predictions of our model and observational measurements of freeze-out temperature fluctuations and primordial abundances of ${}^{4}He$ and *D*, we constrain the free κ-parameter, discussing to what extent the Kaniadakis framework can provide a successful description of the observed Universe.

## 1. Introduction

In the last several decades, several approaches of statistical mechanics have been used in high energy physics to analyze cosmological models [1], particle interactions [2], Lorentz-violating extensions of the Standard Model [3], black holes and other gravitational systems [4,5,6]. Despite the different contexts, a common thread among all of these studies is the adoption of Boltzmann–Gibbs–Shannon (BGS) entropy, which conducts the celebrated Maxwell–Boltzmann exponential distribution according to the Jaynes maximum entropy principle. However, it is well-known that Boltzmann–Gibbs formalism exhibits severe restrictions when applied to many complex systems, such as out-of-equilibrium, long-interacting and thermally fluctuating systems [7]. This motivates the introduction of a more general setting that contains the Maxwell–Boltzmann distribution measure as a special case.

Among the most popular generalizations, evidence from relativistic particle systems [8,9] has suggested a non-exponential distribution function with power tails that originated from *Kaniadakis entropy* [10,11,12]
(1)$$S_{\kappa }\;=\;-\sum_{i}n_{i}\ln_{\kappa }n_{i}\;$$
where the κ-deformed logarithm is defined by (here, and henceforth, we use natural units $k_{B}=\hbar =G=c=1$:)
(2)$$\ln_{\kappa }x\;\equiv \;\frac{x^{\kappa }-x^{-\kappa }}{2\kappa }\;\;-1<\kappa <1\;$$
and the generalized Boltzmann factor for the *i*-th level of the system takes the form
(3)$$n_{i}\;=\;\alpha \exp_{\kappa }\left[-\beta \left(E_{i}-\mu \right)\right].$$
Here, the κ-deformed exponential is given by
(4)$$\exp_{\kappa }\left(x\right)\;\equiv \;{\left(\sqrt{1+\kappa^{2}\;x^{2}}\;+\;\kappa \;x\right)}^{1/\kappa }\;$$
while
(5)$$\alpha \;=\;{\left[(1-\kappa )/(1+\kappa )\right]}^{1/2\kappa }\;\;1/\beta \;=\;\sqrt{1-\kappa^{2}}\;\;T\;.$$

As shown in [11], the statistical model developed from the generalized functions (Equation 2) and (Equation 4) emerges naturally and unequivocally within the framework of special relativity. Notice that the κ parameter is not fixed by the theory and should be constrained via theoretical and/or observational analyses. It is straightforward to check that the classical κ→0 limit reproduces the ordinary (Maxwell–Boltzmann) statistical mechanics, thus making Equation (Equation 1) a self-consistent relativistic generalization of BGS entropy formula.

By using the definition of microcanonical ensemble, it has been argued that, for the case of black holes, the κ-entropy (Equation 1) can be rewritten in the form [13,14]
(6)$$S_{\kappa }\;=\;\frac{1}{\kappa }\sinh\left(\kappa \;S_{BH}\right)\;$$
where $S_{BH}=A/\left(4\right)$ is the standard Bekenstein–Hawking entropy, which is still recovered for κ→0. Since $S_{\kappa }=S_{-\kappa }$, in what follows, we can restrict to the κ≥0 case without a loss of generality.

While being expressly formulated for black holes, Equation (Equation 6) can also be used within the cosmological framework in the lines of gravity-thermodynamic conjecture. For instance, in [15], Drepanou et al. have shown that Holographic Dark Energy based on Kaniadakis entropy (Equation 6) leads to interesting cosmological behavior, retracing the standard thermal history of the Universe in good agreement with observations [16,17]. Similarly, in [18,19] (and references therein), Kaniadakis statistics has been used in gravity scenarios to address the Jeans instability and simulate dark matter-like effects, respectively. Applications of κ-entropy can also be found in plasma physics, astrophysics, information theory, fluid dynamics and other fields (see [20] for a recent review). All of this makes Kaniadakis statistics a very flexible framework that can potentially adapt to the diversity of relativistic physical contexts where Maxwell–Boltzmann distribution fails, thus motivating a careful analysis of the subject.

In passing, we mention that, besides Kaniadakis formulation, there exist many other generalized entropies which are relevant and commonly used in physics, for instance Tsallis, Abe, Landsberg–Vedral, Sharma–Mittal, Rény and Barrow entropies, among others (see [21] for a detailed review), all containing the classical BGS entropy as a special case. While exhibiting some mathematical similarities with κ-deformation, these entropy measures are better suited to describe statistical properties of long-interacting, dissipative or large-scale fluctuating systems, but do not work properly in the relativistic realm, which is the framework of this review. Clearly, one could in principle extend the present study to the above family of generalized entropies to see how temperature fluctuations, non-extensive (Tsallis-like) or quantum gravity (Barrow-like) corrections affect ensuing scenarios in comparison with Kaniadakis conjecture. Some of these research lines have already been explored in literature [22,23,24,25,26]. However, a systematic investigation of these aspects goes beyond the scope of the present review and will be considered elsewhere.

Starting from the above premises, in the present manuscript, we focus on the study of Kaniadakis statistics applied to Gravity and Cosmology. The structure of the work can be basically divided into two parts: the first one (Section 2) contains a review of recent advances in the literature. Special care is devoted to examine:implications on open stellar clusters: such systems are physically related group of stars held together by the mutual gravitational attraction. Since their constituents typically have similar age and chemical composition, they provide very important laboratories where stellar properties compared to isolated field stars can be studied. In [22], Carvalho et al. have shown that the characteristic relaxation mechanism associated with radial orbital instability cannot be understood in the classical Maxwell–Boltzmann framework, emphasizing the need of non-Gaussian (Kaniadakis-like) statistics to fit the distribution of stellar residual radial velocity in some baseline stellar open clusters.Jeans instability and gravitational collapse: the dynamical stability of a self-gravitating system can be described by the Jeans criterion, which states that, if the wavelength of a density fluctuation inside the system is greater than a certain threshold given by the Jeans wavelength, then the density will grow in time exponentially, and the system becomes gravitationally unstable. In [27], this criterion has been revisited in the context of Kaniadakis statistics, obtaining a κ-deformed critical wavelength larger than the standard expression. Similar studies have also been developed in Eddington-inspired Born–Infeld [28] and f(R) [29] gravity, and the dark-baryonic matter model [30], among others.Holographic Cosmology: Holographic Dark Energy (HDE) is a theoretical framework that arises from the attempt of applying the holographic principle to the dark energy problem [31]. A crucial ingredient in the construction of this model is the relationship between the entropy of the Universe (conceived as a thermodynamic system) and its geometrical properties, such as its radius. The standard HDE scenario is built upon Bekenstein–Hawking entropy, which arises as the black hole application of the BGS one. However, in [13,14,15,32], a generalized HDE based on Kaniadakis entropy has been investigated along with its implications on the cosmic evolution and thermal properties of the Universe (see also [23,24,33,34,35] for further applications). Remarkably, it has been shown that Kaniadakis dark energy exhibits peculiar features that do not have any correspondence in the traditional HDE, potentially providing a way to alleviate the Hubble tension.Entropic gravity: Verlinde’s conjecture of entropic gravity [36] presents gravitational force as an emergent (rather than fundamental) force caused by changes in the information associated with the positions of material bodies. Starting from this idea, an effective gravitational constant can be derived and used to introduce Kaniadakis statistics, the ensuing method being a simpler alternative to the usual procedure employed in non-Gaussian statistics. In [25], such a formalism has been applied to infer Kaniadakis-induced corrections to the Jeans criterion for self-gravitating systems, as well as to establish a connection with deviations of Newton’s law arising in a submillimeter range (for the sake of transparency, it must be said that the issue of whether Newton’s law exhibits deviations from inverse square behavior in submillimeter regime is quite controversial. For instance, in [37], it has been found that there are still no deviations in separation down to $O(10^{2})$ μm.) [38].Black hole thermodynamics: inspired by a dual Rény entropy [39,40], in [41], Abreu et al. have suggested and applied a dual Kaniadakis entropy to black hole thermodynamics. In this way, a generalized equipartition theorem has been derived, leading to a κ-modified black hole temperature and heat capacity. In addition, it has been argued that black holes in Kaniadakis statistics could exhibit a thermally stable phase, thus opening new glimpses into the study of black hole thermodynamics at both theoretical and phenomenological levels [41].Loop Quantum Gravity (LQG): this is a well-known non-perturbative and background independent theory of gravity which aims to merge Quantum Mechanics and General Relativity [42]. One of the characteristic parameters of LQG is the so-called Immirzi parameter [43], which is an arbitrary number that measures the size of the quantum of area in Planck units. By using Kaniadakis statistics, Abreu et al. have derived a non-trivial relation between the Immirzi parameter, the κ deformation parameter and the area of a punctured surface [44], which is a topological two-sphere with defects carrying spin quantum numbers endowed by the edges of the spin network that represents the bulk quantum geometry. The question arises as to whether Kaniadakis statistics might play any role in the context of quantum gravity.

On the other hand, the second part of the manuscript (Section 3) provides the original contribution of this work. Here, we study consequences of Kaniadakis Holographic Dark Energy (KHDE) on Big Bang Nucleosynthesis (BBN). Specifically, we constrain the deformation κ-parameter by requiring consistency between theoretical predictions of our model and observational data of primordial abundances and freeze-out temperature fluctuations, which only allow for very tiny deviations from General Relativity. Finally, in Section 4, we summarize results and discuss some possible future challenges of Kaniadakis statistics aimed at both broadening the current research lines and opening novel prospects in this field.

## 2. Gravity and Cosmology in Kaniadakis Statistical Theory: Recent Advances

In this section, we discuss some recent findings pertaining to Gravity and Cosmology in Kaniadakis statistics. Our aim is to highlight the advantages of Kaniadakis model in describing some phenomena which are not well framed (or not even understood) in Maxwell–Boltzmann theory.

### 2.1. Open Stellar Clusters

Open stellar clusters are a type of star cluster made of up to a few thousand stars that have roughly the same age and composition, being formed from the same giant molecular cloud. The knowledge of the different properties of these clusters, such as the distribution of dispersion velocity and phase density, is needed to establish the statistical laws and the relaxation mechanisms that rule their evolution.

There are essentially three main mechanisms: collisional relaxation, which is characterized by a Maxwell-like distribution; the Lynden–Bell relaxation, leading to a Fermi distribution; and a relaxation associated with radial orbit instability that attains a non-monotonic distribution. While the first two mechanisms are well described by the standard statistical mechanics, the last one is not well-understood yet.

The above issue has been examined in [22] in the background of Kaniadakis statisticsSpecifically, Carvalho et al. have investigated the effects of non-Gaussianity on the distribution of stellar residual radial velocity in some open clusters’ samples. The generalized κ-distribution function they find for the radial velocity $v_{r}$ has the form
(7)$$\begin{array}{ccc} \phi_{\kappa }\left(v_{r}\right) & = & A_{\kappa }{\left[\sqrt{1+\kappa^{2}{\left(\frac{v_{r}^{2}}{\sigma_{\kappa }^{2}}\right)}^{2}}-\kappa \frac{v_{r}^{2}}{\sigma_{\kappa }^{2}}\right]}^{1/\kappa }\\ & = & A_{\kappa }\;\exp_{\kappa }\left(-\frac{v_{r}^{2}}{\sigma^{2}}\right), \end{array}$$
where $A_{\kappa }$ is a constant, and $\sigma_{\kappa }$ denotes the characteristic distribution width. In the second step, we have used the definition (Equation 4). For κ→0, this expression reduces to the standard Gaussian distribution
(8)$$\phi \left(v_{r}\right)\;=\;A\;\exp\left(-\frac{v_{r}^{2}}{\sigma^{2}}\right),$$
as expected.

By using the Kolmogorov–Smirnov statistical test, the best $\phi_{\kappa }\left(v_{r}\right)$ has been obtained for each observed cumulative distribution of the residual radial velocities. As a result, it has been shown that the generalized Kaniadakis distribution fits data much better than the standard Gaussian does, provided that one allows the κ parameter to be varying with the stellar-cluster ages. Below, we will show that a similar running behavior is supported by completely independent arguments in the framework of Kaniadakis Cosmology.

### 2.2. Jeans Instability and Gravitational Collapse

Jeans criterion provides a condition to establish whether a self-gravitating system is stable under the effects of its internal gas pressure. The so-called Jeans length that represents the watershed between stable and unstable systems is given by [45]
(9)$$\lambda_{J}=\sqrt{\frac{\pi T}{\mu m_{H}\rho_{0}}}\;$$
where *T* is the temperature, μ the mean molecular weight, $m_{H}$ the hydrogen atomic mass and $\rho_{0}$ the equilibrium mass density of the system, respectively.

According to Jeans instability, if the wavelength λ of a density perturbation is higher than $\lambda_{J}$, then the density grows exponentially, giving rise to an unstable system. Otherwise, stability is kept. The same criterion can also be expressed in terms of a critical mass for self-gravitating systems (see Section 2.4).

The condition (Equation 9) follows from the canonical equipartition theorem in Maxwell–Boltzmann statistics. Nevertheless, motivated by relativistic considerations, in [27], Abreu et al. have shown that the critical density $\lambda_{J}$ gets non-trivially modified in the context of Kaniadakis statistics. In particular, by using the κ-generalized equipartition theorem
(10)$$E_{\kappa }\;=\;\frac{1}{2}N\;\frac{\left(1+\frac{\kappa }{2}\right)}{\left(1+\frac{3}{2}\kappa \right)2\kappa }\;\;\frac{\Gamma \left(\frac{1}{2\kappa }-\frac{3}{4}\right)\Gamma \left(\frac{1}{2\kappa }+\frac{1}{4}\right)}{\Gamma \left(\frac{1}{2\kappa }+\frac{3}{4}\right)\Gamma \left(\frac{1}{2\kappa }-\frac{1}{4}\right)}\;T\;$$
and Verlinde formalism of entropic gravity [36], the following expression for the κ-deformed critical wavelength has been exhibited
(11)$$\lambda_{c}^{\kappa }\;=\;\sqrt{\frac{\left(1+\frac{\kappa }{2}\right)}{\left(1+\frac{3}{2}\kappa \right)2\kappa }\;\;\frac{\Gamma \left(\frac{1}{2\kappa }-\frac{3}{4}\right)\Gamma \left(\frac{1}{2\kappa }+\frac{1}{4}\right)}{\Gamma \left(\frac{1}{2\kappa }+\frac{3}{4}\right)\Gamma \left(\frac{1}{2\kappa }-\frac{1}{4}\right)}}\;\lambda_{J}\;$$
where Γ is the Gamma function. From this equation, we infer that Jeans instability is modified in such a way that:-for $\kappa =0⟹\lambda_{c}^{\kappa }=\lambda_{J}$, i.e., the classical criterion is restored;-for $0<\kappa <2/3⟹\lambda_{c}^{\kappa }>\lambda_{J}$;-for $\kappa \to 2/3^{-}⟹\lambda_{c}^{\kappa }\to \infty$, which means that the self-gravitating systems are always stable (notice that for κ≥2/3 the modified equipartition law based on Kaniadakis statistics diverges, thus making the derivation of the generalized Jeans criterion meaningless.).

Apart from the limit case κ=0, we then see that the modified critical wavelength is always larger than the corresponding Maxwell–Boltzmann value. In other terms, Kaniadakis statistics predicts self-gravitating systems to be more stable compared to the classical scenario.

Furthermore, one can show that the κ-deformed entropy also affects the physical temperature of gravitating systems and the velocity of the propagation of sound inside them. The resulting expressions are [27]
(12)$$T_{\kappa }=\frac{\left(1+\frac{\kappa }{2}\right)}{\left(1+\frac{3}{2}\kappa \right)2\kappa }\;\frac{\Gamma \left(\frac{1}{2\kappa }-\frac{3}{4}\right)\Gamma \left(\frac{1}{2\kappa }+\frac{1}{4}\right)}{\Gamma \left(\frac{1}{2\kappa }+\frac{3}{4}\right)\Gamma \left(\frac{1}{2\kappa }-\frac{1}{4}\right)}\;T\;\;v_{\kappa }^{s}=\sqrt{\frac{T_{\kappa }}{m}}\;.$$

These results have been tested considering 16 galaxy clusters. It has been found that Boltzmann–Gibbs statistics are consistent with data, although non-Gaussian effects cannot be completely ruled out, constraining 0≤κ≤0.034 at the 2σ confidence level.

It is worth mentioning that Jeans criterion in Kaniadakis statistics has also been addressed in other different contexts. For instance, in [28], Jeans instability has been revisited in the framework of Eddington-inspired Born–Infeld gravity, showing that the κ-deformed distribution may have non-negligible effects on the Jeans modes of the collisionless Eddington-inspired Born–Infeld gravitational systems. In a similar fashion, the influence of the κ-generalized Jeans criterion has been examined in [29] in f(R) gravity for both high and low frequency density perturbations. As a result, it has been proven that the range of the unstable modes and the growth rates decrease with increasing values of κ. A further step forward has been taken in [30], where implications of Kaniadakis statistics have been explored on gravitational systems composed also by dark and baryonic matter. The analysis of the κ-modified dispersion equation for such systems has pointed out that Jeans instability is suppressed comparing to the standard case, implying that Kaniadakis corrections oppose the gravitational collapse. Clearly, all of the above outcomes disclose a new class of phenomena and/or mechanisms, which potentially allow us to highlight signatures of Kaniadakis statistics in gravitational systems.

### 2.3. Holographic Dark Energy

Holographic Dark Energy is a dynamical model of dark energy built on the usage of the holographic principle and Bekenstein–Hawking area law. Although cosmological applications of HDE have been extensively considered in the past literature [46,47,48], its shortcomings in reproducing the thermal history of the Universe have motivated some tentative changes over the years [14,15,49,50,51,52,53]. Among these generalizations, promising results have been provided by HDE based on Kaniadakis entropy (Kaniadakis Holographic Dark Energy, KHDE). Several models of KHDE have been proposed: here, we refer to the approach of [14], which correctly reduces to the usual HDE in the κ→0 limit and does not involve large deviations from standard entropy to describe the Universe evolution, as it should be according to Equation (Equation 2). For other possible extensions, see also [13,32,33].

Starting from the entropy (Equation 6) and using the gravity-thermodynamic conjecture, in [14], Lymperis et al. have derived modified Friedmann equations ruling the evolution of a homogeneous and isotropic Friedmann–Robertson–Walker (FRW) geometry filled with matter and dark energy fluids and bounded by the apparent horizon (see Section 3). In turn, these equations allow for computing characteristic cosmic parameters, such as the Equation of State (EoS) parameter, the deceleration parameter, the squared speed of sound and the Hubble parameter, to be compared with the theoretical predictions of the Λ-CDM model of Cosmology. A more detailed experimental analysis of KHDE has been carried out in [15,16,17], showing that KHDE predictions do agree with observational data and might contribute to alleviate the $H_{0}$ tension too. In particular, concerning the dark energy EoS parameter, it has been found to exhibit a phenomenology richer than standard HDE, being quintessence-like, phantom-like, or experiencing the phantom-divide crossing in the past or in the future. In addition, observations from Supernovae type Ia and Baryon Acoustic Oscillations data enable constraining the Kaniadakis parameter around the vanishing value, consistently with the expectation of small deviations from Gaussian-like statistics in nature.

### 2.4. Entropic Gravity

In the entropic gravity formalism by Verlinde [36], gravitational force is conceived as an entropic force caused by changes in the information associated with the positions of material bodies. This conjecture combined with the generalized Kaniadakis’ equipartition law gives an effective gravitational constant in the form [25]
(13)$$G_{\kappa }\;=\;\frac{\left(1+\frac{3}{2}\kappa \right)2\kappa }{\left(1+\frac{\kappa }{2}\right)}\;\frac{\Gamma \left(\frac{1}{2\kappa }+\frac{3}{4}\right)\Gamma \left(\frac{1}{2\kappa }-\frac{1}{4}\right)}{\Gamma \left(\frac{1}{2\kappa }-\frac{3}{4}\right)\Gamma \left(\frac{1}{2\kappa }+\frac{1}{4}\right)}G\;$$
where here we have restored the gravitational constant *G* for the sake of clarity.

In the same spirit as Section 2.2, Equation (Equation 13) can be used to describe Kaniadakis statistics effects on Jeans mass criterion in self-gravitating systems. Specifically, Abreu et al. have derived an expression for the modified Jeans critical mass as [25]
(14)$$M_{J}^{\kappa }\;=\;{\left[\frac{\left(1+\frac{\kappa }{2}\right)}{\left(1+\frac{3}{2}\kappa \right)2\kappa }\;\frac{\Gamma \left(\frac{1}{2\kappa }-\frac{3}{4}\right)\Gamma \left(\frac{1}{2\kappa }+\frac{1}{4}\right)}{\Gamma \left(\frac{1}{2\kappa }+\frac{3}{4}\right)\Gamma \left(\frac{1}{2\kappa }-\frac{1}{4}\right)}\right]}^{3/2}\;M_{J},$$
where
(15)$$M_{J}={\left(\frac{5\;T}{m}\right)}^{3/2}{\left(\frac{3}{4\pi \rho }\right)}^{1/2}\;$$
is the usual Jeans mass in Maxwell–Boltzmann framework, while all other quantities are defined as in Section 2.2. As before, one can distinguish three possible regimes given by: (*i*) κ=0, which implies instability for $M>M_{J}^{\kappa }=M_{J}$; (*ii*) 0<κ<2/3, which entails $M>M_{J}^{\kappa }>M_{J}$; and (*iii*) $\kappa \to 2/3^{-}$, in which $M_{J}^{\kappa }\to \infty$, giving rise to an always stable gravitational system. This confirms the previous result that Kaniadakis entropy opposes gravitational collapse.

Another important physical quantity considered in [25] is the free fall time
(16)$$t_{FF}=\sqrt{\frac{3}{2\pi \rho }},$$
which is defined as the time necessary to the system to finally collapse. In the case of Kaniadakis statistics, this turns out to be modified as [25]
(17)$$t_{FF}^{\kappa }\;=\;{\left[\frac{\left(1+\frac{\kappa }{2}\right)}{\left(1+\frac{3}{2}\kappa \right)2\kappa }\;\frac{\Gamma \left(\frac{1}{2\kappa }-\frac{3}{4}\right)\Gamma \left(\frac{1}{2\kappa }+\frac{1}{4}\right)}{\Gamma \left(\frac{1}{2\kappa }+\frac{3}{4}\right)\Gamma \left(\frac{1}{2\kappa }-\frac{1}{4}\right)}\right]}^{1/2}t_{FF}\;.$$
which indicates that $t_{FF}^{\kappa }>t_{FF}$, except for the κ=0 case, where equality is recovered. Thus, the self-gravitating system in Kaniadakis scenario takes more time to collapse comparing to the Gaussian framework.

As a further application, in [25], a connection has been studied between the modifications of Newton’s law induced by the generalized gravitational constant (Equation 13) and possible deviations arising in a submillimeter range, which are parameterized by [38]
(18)$$G\left(r\right)\;=\;G\left[1+\alpha \left(1+\frac{r}{\lambda }\right)e^{-r/\lambda }\right],$$
where α is a dimensionless parameter and λ gives the energy (or length) scale at which departures of Newton’s law from the standard inverse square behavior should occur.

In so doing, Abreu et al. have obtained a relationship between α and the Kaniadakis parameter in the form
(19)$$\alpha \;=\;\frac{\left(1+\frac{3}{2}\kappa \right)2\kappa }{\left(1+\frac{\kappa }{2}\right)}\;\frac{\Gamma \left(\frac{1}{2\kappa }+\frac{3}{4}\right)\Gamma \left(\frac{1}{2\kappa }-\frac{1}{4}\right)}{\Gamma \left(\frac{1}{2\kappa }-\frac{3}{4}\right)\Gamma \left(\frac{1}{2\kappa }+\frac{1}{4}\right)}-1\;.$$

This result opens up the tantalizing possibility of probing Kaniadakis-like deviations from Gaussianity via tests of the gravitational inverse square law in experimentally accessible regions.

### 2.5. Black Hole Thermodynamics

Inspired by a novel type of Rény entropy proposed in [40], Abreu et al. have introduced a dual Kaniadakis entropy in the form [41]
(20)$$\begin{array}{ccc} S_{\kappa }^{*} & = & \frac{1}{\kappa }\log\left(\kappa \;S_{BH}+\sqrt{1+\kappa^{2}\;S_{BH}^{2}}\right)\\ & = & \log\left[\exp_{\kappa }\left(S_{BH}\right)\right], \end{array}$$
where $S_{BH}$ is the Bekenstein–Hawking entropy defined below Equation (Equation 6).

Based on the above equation, one can derive a modified Hawking temperature and heat capacity of black holes as [41]
(21)$$\begin{array}{ccc} T & = & \frac{\sqrt{1+16\kappa^{2}\pi^{2}M^{4}}}{8\pi M}\; \end{array}$$
(22)$$\begin{array}{ccc} C_{BH} & = & -\frac{8\pi M^{2}{\left(1+16\kappa^{2}\pi^{2}M^{4}\right)}^{1/2}}{1-16\kappa^{2}\pi^{2}\;M^{4}}\; \end{array}$$
where *M* denotes the black hole mass. Notice that both of the above expressions reproduce the semiclassical Hawking results
(23)$$T=1/\left(8\pi M\right)\;\;C_{BH}=-8\pi M^{2}\;$$
in the κ→0 limit, as it should be.

Remarkably, from Equation (Equation 22), we see that $C_{BH}$ takes negative (positive) values for $M<{\left[2{\left(\kappa \pi \right)}^{1/2}\right]}^{-1}$ ($M>{\left[2{\left(\kappa \pi \right)}^{1/2}\right]}^{-1}$), leading to a thermally unstable (stable) black hole. Such a result implies that it is possible for a phase transition to occur in the dual Kaniadakis statistics framework—a result which has no correspondence in the Maxwell–Boltzmann scenario. This points out the potential relevance of dual Kaniadakis entropy in the analysis of black hole thermodynamics.

### 2.6. Loop Quantum Gravity

Implications of Kaniadakis statistics have also been analyzed in Loop Quantum Gravity, which arises from the effort to grasp what quantum spacetime is at the fundamental level. More specifically, this formalism is characterized by quantum operators for areas and volumes that exhibit discrete spectra.

One of the peculiar parameters of LQG is the so-called Immirzi parameter, which is a free dimensionless quantity that provides the size of a quantum of area in Planck units. A way to compute this parameter is by counting the number of microstates of a given system in LQG. For black holes, this is typically accomplished by use of the Bekenstein–Hawking entropy area law, which roots its origin in the BGS entropy. As a result, one obtains [43]:(24)$$\gamma =\frac{\log2}{\pi \sqrt{3}}\;.$$

This expression can be straightforwardly generalized to the background of Kaniadakis statistics by using the κ-deformed entropy in the microcanonical ensemble. Calculations have been carried out in [44] for a generic system of surface area *A*, leading to
(25)$$\begin{array}{ccc} \gamma_{\kappa } & = & \gamma \;\frac{\frac{\kappa A}{4}}{\log\left[\frac{\kappa A}{4}+\sqrt{1+\frac{\kappa^{2}A^{2}}{16}}\right]}\\ & = & \gamma \frac{\frac{A}{4}}{\log\left[\exp_{\kappa }\left(\frac{A}{4}\right)\right]}\; \end{array}$$
which correctly reduces to γ for κ→0. Since the extra factor appearing in the above relation is greater than unity, we have $\gamma_{\kappa }>\gamma$, resulting in a larger size of the quantum of area in Planck units.

The outlined κ-dependence of the Immirzi parameter reveals a non-trivial interplay between LQG and Kaniadakis statistics. Hence, much effort is needed to better understand the potential rôle of Kaniadakis entropy within the framework of quantum gravity.

## 3. Big Bang Nucleosynthesis in Kaniadakis Statistics

In physical Cosmology, Big Bang Nucleosynthesis (BBN) refers to the sequence of nuclear processes which synthesized primordial light elements, such as Hydrogen *H*, Deuterium *D*, Helium isotopes ${}^{3}He$ and ${}^{4}He$ and Lithium isotope ${}^{7}Li$ [54]. Clearly, since BBN drives the whole evolution of Universe’s chemical composition, BBN parameters must be very tightly constrained to be consistent with current observations. Therefore, this phenomenon provides an unparalleled arena to test cosmological models and constrain related parameters with great accuracy.

An interesting issue to address is how relativistic degrees of freedom of the early Universe affect BBN when described in the more proper framework of Kaniadakis statistics. To solve the problem, let us first derive modified Friedmann equations in Kaniadakis Cosmology. For this purpose, we mainly refer to [14,15], though we feature κ-induced corrections in a slightly different way.

Consider a homogeneous and isotropic FRW flat geometry of metric
(26)$$ds^{2}\;=\;-dt^{2}+a^{2}\left(t\right)\left(dr^{2}+r^{2}d\Omega^{2}\right),$$
where a(t) is the scale factor and *t* the cosmic time. In addition, we assume that the Universe is filled up with with a matter perfect fluid of equilibrium mass density $\rho_{0}$ and pressure $p_{0}=w\rho_{0}$, where −1≤w≤1/3 is the EoS parameter.

Invoking the gravity-thermodynamic conjecture, we can think of our Universe as a thermodynamic system bounded by an apparent horizon $r_{a}=1/H=a/\dot{a}$ and endowed with a temperature and entropy obeying the same rules as for black holes (in our notation, the dot indicates a derivative with respect to the cosmic time) [55]. In this scenario, by using the energy–momentum tensor of matter content, the continuity equation and BH entropy area law, we are led to the canonical Friedmann equations
(27)$$\begin{array}{ccc} -4\pi \;\left(\rho_{0}+p_{0}\right) & = & \dot{H}\; \end{array}$$
(28)$$\begin{array}{ccc} \frac{8\pi \;}{3}\rho_{0} & = & H^{2}\; \end{array}$$
where cosmological constant effects have been neglected.

The question now arises as how Equations (Equation 27) and (Equation 28) get modified when using the κ-deformed entropy (Equation 6) instead of BH entropy. Following the same recipe as above, one arrives to the Kaniadakis–Friedmann equations [14]
(29)$$\begin{array}{ccc} -4\pi \left(\rho +p\right) & = & \cosh\left(\kappa \;\frac{\pi }{H^{2}}\right)\dot{H}\; \end{array}$$
(30)$$\begin{array}{ccc} \frac{8\pi }{3}\rho & = & \cosh\left(\kappa \;\frac{\pi }{H^{2}}\right)H^{2}-\kappa \pi \;\mathrm{shi}\left(\kappa \;\frac{\pi }{H^{2}}\right), \end{array}$$
where ρ and *p* now denote the total energy density and pressure including Kaniadakis corrections, and
(31)$$\mathrm{shi}\left(x\right)\;\equiv \;\int_{0}^{x}\frac{\sinh(x^{\prime })}{x^{\prime }}\;dx^{\prime }\;.$$
Notice that the classical scenario is easily recovered for κ→0.

To make the κ-dependence of the l.h.s. in Equations (Equation 29) and (Equation 30) explicit, let us recast the total energy density and pressure as
(32)$$\begin{array}{ccc} \rho & = & \rho_{0}+\delta \rho_{\kappa }\; \end{array}$$
(33)$$\begin{array}{ccc} p & = & p_{0}+\delta p_{\kappa }\; \end{array}$$
where we have separated out Kaniadakis-induced corrections $\delta \rho_{\kappa }$ and $\delta p_{\kappa }$, respectively.

We then expand cosh(x) and shi(x) in Equations (Equation 29) and (Equation 30) for small κ, which is indeed the case, since departures from Maxwell–Boltzmann statistics are expected to be small. We obtain the leading order
(34)$$\begin{array}{ccc} -4\pi \left(\rho_{0}+p_{0}+\delta \rho_{\kappa }+\delta p_{\kappa }\right) & ≃ & \dot{H}\left(1+\frac{\pi^{2}}{2}\frac{\kappa^{2}}{H^{4}}\right), \end{array}$$
(35)$$\begin{array}{ccc} \frac{8\pi }{3}\left(\rho_{0}+\delta \rho_{\kappa }\right) & ≃ & H^{2}-\frac{\pi^{2}}{2}\frac{\kappa^{2}}{H^{2}}\; \end{array}$$
which gives [56]
(36)$$\begin{array}{ccc} \delta \rho_{\kappa } & ≃ & -\frac{9}{128}\frac{\kappa^{2}}{\rho_{0}}, \end{array}$$
(37)$$\begin{array}{ccc} \delta p_{\kappa } & ≃ & \frac{21}{128}\frac{\kappa^{2}}{\rho_{0}}\;. \end{array}$$

For our next purposes, it is now convenient to rewrite the modified Friedmann Equation (Equation 35) in the equivalent form
(38)$$H\left(\rho \right)\;\equiv \;H\left(\rho_{0}\right)\;Z_{\kappa }\left(\rho \right)\;$$
where $H(\rho_{0})$ is the unmodified Hubble parameter obeying Equation (Equation 28) and
(39)$$Z_{\kappa }\left(\rho \right)\;\equiv \;1+\frac{9}{256}\;\frac{\kappa^{2}}{\rho^{2}}\;.$$

This can be further manipulated by resorting to Equations (Equation 32) and (Equation 36) and expressing the equilibrium energy density $\rho_{0}$ as a function of the temperature according to
(40)$$\rho_{0}\left(T\right)=\frac{\pi^{2}\;g\left(T\right)}{30}\;T^{4}\;$$
where g(T) denotes the effective number of degrees of freedom of the Universe at temperature *T*. Since in the following we shall focus on the radiation dominated epoch, we can roughly set g(T)≃10. In so doing, we obtain
(41)$$H\left(\rho \right)\to H\left(T\right)\;=\;\frac{2\pi }{3}\sqrt{\frac{\pi \;g(T)}{5}}\;T^{2}\;\;Z_{\kappa }\left(T\right)\;$$
where
(42)$$Z_{\kappa }\left(T\right)\;\approx \;1+\frac{2025}{64\;\pi^{4}}\;\frac{\kappa^{2}}{g^{2}\left(T\right)\;T^{8}}\;.$$

Before going further, we point out that, in the ordinary Cosmology based on GR and BH entropy, the *Z* function takes unit value, as it can be seen by considering the κ→0 limit. In general, departures of Z−1 from zero could emerge from either extended formulations of gravity, including alternative geometric frameworks and/or different entropic scenarios, or by introducing extra particle degrees of freedom in the standard theory. Since in the present work we are interested in effects induced by Kaniadakis statistics, we focus on the first setting, neglecting corrections brought about by exotic particles. A similar analysis has been recently proposed in [57,58] in the context of Tsallis statistics and generalizations of the Heisenberg principle induced by gravity, respectively.

### 3.1. Freeze-Out Mechanism

According to the BBN model, the current abundances of the first very light atomic nuclei were already nearly defined few minutes after the initial Big Bang, when the energy and number density were still dominated by relativistic degrees of freedom—leptons and photons [59]. Owing to their continuous and repeated collisions, such particles were in thermal equilibrium. In turn, protons and neutrons were kept in equilibrium through the reactions
(43a)$$\left(a\right)\;\;\;\;\nu_{e}+n\;\;⟷\;\;p+e^{-}\;$$
(43b)$$\left(b\right)\;\;\;\;e^{+}+n\;\;⟷\;\;p+{\bar{\nu }}_{e}\;$$
(43c)$$\left(c\right)\;\;\;\;n\;\;⟷\;\;p+e^{-}+{\bar{\nu }}_{e}\;.$$

In this scenario, neutron abundance can be estimated by computing the rate of conversion $\lambda_{pn}\left(T\right)$ of protons into neutrons and its inverse
(44)$$\lambda_{np}\left(T\right)=e^{-Q/T}\lambda_{pn}\left(T\right)\;$$
where $Q=m_{n}-m_{p}≃1.29\;\mathrm{MeV}$, with $m_{n(p)}$ being the neutron (proton) mass. The rate $\lambda_{np}\left(T\right)$ is given by the sum of the three rates for the processes (a), (b) and (c), i.e.,
(45)$$\lambda_{np}\left(T\right)=\lambda_{a}\left(T\right)+\lambda_{b}\left(T\right)+\lambda_{c}\left(T\right)\;.$$

The reactions (43) went on until the decreasing temperature and density content of the Universe caused them to become too slow, at about the *freeze-out* temperature $T_{0f}≃0.6\;\mathrm{MeV}$. In compliance with [59], we require that *T* during the freeze-out period was low enough compared to the typical energy scale for the processes (43). In addition, we assume to neglect the electron mass $m_{e}$ with respect to the electron and neutrino energies. Under these hypotheses, one obtains [59]
(46)$$\lambda_{a}\left(T\right)≃qT^{5}+O\left(\frac{Q}{T}\right)=\lambda_{b}\left(T\right)\;$$
where $q≃10^{-10}\;{\mathrm{GeV}}^{-4}$. On the other hand, $\lambda_{c}\left(T\right)$ is roughly three orders of magnitude lower than $\lambda_{a}\left(T\right)$ and can in principle be neglected.

Let us now observe that the ${}^{4}He$ mass fraction of the total baryonic mass is [60]
(47)$$Y_{p}\equiv \gamma \;\frac{2\;x(t_{f})}{1+x(t_{f})}\;$$
where
(48)$$\gamma =e^{-(t_{n}-t_{f})/\tau }≃1.$$
Here, $t_{f}≃1\;s$ and $t_{n}≃20\;s$ are the freeze-out and nucleosynthesis times, respectively, while τ≃877s is the neutron mean lifetime. Moreover, we have denoted the neutron-to-proton equilibrium ratio by $x\left(t_{f}\right)=e^{-Q/T(t_{f})}$.

Fluctuations of $Y_{p}$ are related to variations of the freeze-out temperature $\delta T_{F}$ by (see [57] and references therein)
(49)$$\delta Y_{p}=Y_{p}\left[\left(1-\frac{Y_{p}}{2\gamma }\right)\log\left(\frac{2\gamma }{Y_{p}}-1\right)-\frac{2t_{f}}{\tau }\right]\frac{\delta T_{f}}{T_{f}}\;.$$

Observational measurements from ${}^{4}He$ emission lines in extragalactic HII regions enable estimating [61]
(50)$$Y_{p}=0.2449\;\;\;\;\left|\delta Y_{p}\right|≲10^{-4}\;.$$

By plugging these values into Equation (Equation 49) and solving for $\delta T_{f}$, we are led to the following variation of the freeze-out temperature
(51)$$\left|\frac{\delta T_{f}}{T_{f}}\right|≲10^{-4}\;.$$

We now have all the ingredients to constrain the Kaniadakis parameter. Indeed, following [57], we can evaluate the freeze-out temperature in the Kaniadakis framework by imposing that the interaction rate (Equation 45) is of the same order as (or small than) the Hubble rate (Equation 41), and setting $\delta T_{f}=T_{f,\kappa }-T_{0f}=T_{f,\kappa }-0.6\;\mathrm{MeV}$, where we have denoted by $T_{f,\kappa }$ the Kaniadakis-corrected freeze-out temperature. The resulting equation has the form
(52)$$y\;T_{f,\kappa }^{11}\;=\;\kappa^{2}\;x+z\;T_{f,\kappa }^{8}\;$$
where
(53)$$\begin{array}{ccc} y & \equiv & 384\sqrt{5}\pi^{4}\;g^{2}\;q\; \end{array}$$
(54)$$\begin{array}{ccc} x & \equiv & 4050\pi^{3/2}\sqrt{g}\; \end{array}$$
(55)$$\begin{array}{ccc} z & \equiv & 128\pi^{11/2}\;g^{5/2}\;. \end{array}$$

Equation (Equation 52) cannot be solved analytically. However, we can infer an upper bound on the Kaniadakis parameter by resorting to numerical evaluation. In order for $T_{f,\kappa }$ to satisfy Equation (Equation 51), we must have
(56)$$\left|\kappa \right|≲10^{-92}\;$$
which shows that the Kaniadakis parameter must be tightly constrained around zero to be consistent with experimental measurements of freeze-out temperature. More comments on the obtained result can be found at the end of the next subsection.

### 3.2. Primordial Abundances of ${}^{4}He$ and Deuterium *D*

Based on the previous considerations, let us now investigate implications of Kaniadakis statistics on the primordial abundances of Helium isotope ${}^{4}He$ and Deuterium *D*. To this aim, we recall that the sequence of nuclear processes leading to the generation of these elements is
(57)$$\begin{array}{ccc} n+p & \to & D+\gamma \; \end{array}$$
(58)$$\begin{array}{ccc} D+D & \to & {}^{3}He+n\; \end{array}$$
(59)$$\begin{array}{ccc} D+D & \to & T+p\;. \end{array}$$

In the final stage, Deuterium and Tritium *T* or Deuterium and Helium isotope ${}^{3}He$ combine to give
(60)$$\begin{array}{ccc} D+T & \to & {}^{4}He+n\; \end{array}$$
(61)$$\begin{array}{ccc} D+{}^{3}He & \to & {}^{4}He+p\;. \end{array}$$

From [62], we know that the primordial ${}^{4}He$ abundance is constrained by the numerical best fit to the value
(62)$$Y_{p}=0.2485\pm 0.0006+0.0016\left[\left(\eta_{10}-6\right)+100\left(Z-1\right)\right],$$
where the baryon density number is given by $\eta_{10}\equiv 10^{10}\eta_{B}≃6$, with $\eta_{B}$ being the baryon-to-photon ratio [62]. Of course, here we have to consider *Z* equal to Equation (Equation 42) to study Kaniadakis entropy effects on ${}^{4}He$ abundance. For Z=1 (standard GR value), we get back $Y_{p}=0.2485\pm 0.0006$, according to the predictions of the traditional BBN model.

Now, as shown in [57], the requirement of consistency between Equation (Equation 62) and observational measurements of ${}^{4}He$ abundance gives
(63)$$\delta Z\;\equiv \;Z-1≲O(10^{-2})\;.$$
By taking $Z=Z_{\kappa }$, the above equation allows us to constrain the κ deformation parameter to
(64)$$\left|\kappa \right|≲10^{-88}\;$$
where we have considered T≃10MeV.

We can repeat the same considerations as above for the case of *D* abundance. The best numerical fit from [63] gives in this case
(65)$$y_{D_{p}}=2.6\left(1\pm 0.06\right){\left(\frac{6}{\eta_{10}-6\left(Z-1\right)}\right)}^{1.6}\;$$
which still leads to the standard BBN prediction $y_{D_{p}}=2.6\pm 0.16$ for $\eta_{10}=6$ and Z=1. Observational measurements of *D* abundance combined with Equation (Equation 65) set again $\delta Z≲O(10^{-2})$ [57,64], thus leading to the same bound as in Equation (Equation 64).

It is worth noting that the constraint (Equation 64) is less tight than both the bound in Equation (Equation 56) and than results obtained in [16] via cosmic chronometers/Supernovae type Ia ($\left|\kappa \right|≃10^{-124}$) and Baryon Acoustic Oscillations ($\left|\kappa \right|≃10^{-125}$) measurements. Although not contemplated in the original formulation by Kaniadakis, such an apparent incompatibility could be understood by allowing Kaniadakis parameter to be running in time. This scenario would not be surprising in Kaniadakis Cosmology: in fact, it is legitimate to expect that the Universe degrees of freedom encoded by holographic entropy may evolve from an initial description obeying relativistic (Kaniadakis-like) laws to a classical (Boltzmann-like) picture at present time, just as it happens for matter–energy degrees of freedom. In this framework, departures from the standard BGS entropy would be quantified by a time-dependent, or equivalently, temperature-dependent parameter κ≡κ(t)≡κ(T), such that κ substantially differs from zero at high *T* (early stages of the Universe), while it recovers the classical κ→0 behavior as the Universe cools down. This would justify why the bound (Equation 56), which has been derived for the phase of the Universe corresponding to T≃0.6MeV, is more stringent than that in Equation (Equation 64), associated with T=10MeV.

We emphasize that the possibility of a running κ has already been discussed in [56]. Here, we have found further confirmation of this hypothesis. In addition, we point out that a similar proposal has been recently put forward in the context of Tsallis generalized statistics in [65,66,67,68,69], among others.

## 4. Discussion and Conclusions

In this work, we have discussed recent advances of Gravity and Cosmology in Kaniadakis statistical theory. Special care has been devoted to review effects of generalized κ entropy on open stellar clusters, Jeans instability and gravitational collapse, Holographic Dark Energy, Entropic Gravity formalism, Black hole thermodynamics and Loop Quantum Gravity. For each of these frameworks, we have shown that Kaniadakis statistics manifests through non-trivial modifications of characteristic theoretical predictions, such as the distribution of residual radial velocity of open clusters (Equation (Equation 7)) [22], the critical Jeans wavelength/mass (Equations (Equation 11) and (Equation 14)) [27], the gravitational constant (Equation (Equation 13)) and free fall time (Equation (Equation 17)) [25], the black hole temperature and heat capacity (Equation (Equation 21)) [41] and the Immirzi parameter (Equation (Equation 25)) [43]. The ensuing κ-dependent expressions unveil potential mechanisms to test Kaniadakis-induced deviations from Boltzmann statistics in Gravity and Cosmology scenarios.

On the other hand, we have focused on the study of Holographic Dark Energy in Kaniadakis Cosmology. Although this model is well-established in literature at both theoretical and observational levels [13,14,15,16,17], here we have followed an alternative procedure to treat the κ-modified Friedmann equations ruling the evolution of the Universe in Kaniadakis Cosmology (see Equations (Equation 34) and (Equation 35)). These equations have been used to analyze BBN and, in particular, the freeze-out mechanism and the generation of primordial elements. By demanding consistency between theoretical predictions of our model and observational constraints on freeze-out temperature fluctuations and abundances of ${}^{4}He$ and *D*, we have constrained departures from BGS entropy, showing that the κ parameter must be tightly bounded around the vanishing value to be consistent with phenomenology. Remarkably, it has been found that different stages of the Universe evolution correspond to different upper bounds on κ (see Equations (Equation 56) and (Equation 64)). This result opens up the possibility that a realistic description of the history of the Universe in Kaniadakis Cosmology is allowed, provided that one considers a running κ. Clearly, in order to substantiate this paradigm, the above analysis should be carried out by assuming an ab initio time- (or temperature-) dependent κ. This requires further investigation and will be presented elsewhere.

Other aspects are to be explored. Here, we present a list of some possible future challenges:-as a first extension of the above analysis, it would be interesting to search for signatures of inflationary perturbations propagated during the hypothetical Kaniadakis cosmic epoch in present/upcoming experiments on primordial gravitational waves, such as VIRGO, LIGO or LISA. This work is already in progress.-It has been recently argued that Holographic Dark Energy construction might alleviate the $H_{0}$ tension [70], the reason being that it could lead to the phantom regime for dark energy. Since KHDE has been shown to exhibit this feature [15], it is worth going more deeply into the problem to understand whether KHDE may provide a good candidate toward a solution to the $H_{0}$ tension.-In [58], BBN has been studied by using the Generalized Uncertainty Principle (GUP), which emerges from the phenomenological attempt to embed gravity corrections in quantum mechanics so as to predict a minimal length at Planck scale (see [71,72,73] and references therein). Specifically, it has been shown that GUP enters Friedmann equations through a deformation of the entropy area law, which in turn modifies the density/temperature dependence of Hubble constant. Primordial abundances evaluated in this way exhibit a non-trivial dependence on the GUP deformation parameter. Given the formal analogies between such a result and those obtained in the present context, the question naturally arises as to whether any kind of connection between Kaniadakis statistics and GUP can be established at a more fundamental level. We expect that this study could also pave the way toward formulating a relativistic model of GUP.-As argued in Section 2.5, the analysis of black holes thermodynamics from a dual Kaniadakis entropy reveals a possible thermally stable phase in Kaniadakis statistics, a fact that cannot be noticed when working in Boltzmann theory. This shows that κ-deformed entropy not only generalizes standard results, but also predicts features that do not have any correspondence in the ordinary black hole thermodynamics. Without a doubt, a thorough examination of this framework is needed to find out all peculiarities of black holes in Kaniadakis statistics.-In the recent study of [30], effects of Kaniadakis statistics on the Jeans instability have been analyzed for gravitational systems composed by dark and baryonic matter. As a result, it has been found that instability is suppressed in comparison with the background of Maxwell distribution and, thus, opposes the gravitational collapse of such systems. An appealing extension of this work is to consider Kaniadakis implications on Jeans instability of partially ionized dusty plasma and discuss their relevance in the formation of planetesimals and collapse of interstellar clouds in star forming regions.-Based on the quantum tunnelling concept and Boltzmann statistics, one can derive the critical Gamow temperature $T_{G}$ at which the star-burning process occurs. The problem has been recently addressed in the context of Kaniadakis statistics [26], showing that Gamow temperature decreases with respect to the standard value. Therefore, stars whose burning temperature differs from $T_{G}$ might be signals of deviations from Gaussian statistics in stellar sciences. This provides a challenging framework where to test the Kaniadakis theory experimentally.-In [34], Abreu et al. have derived a κ-modified version of the Tully–Fisher relation, which connects the rotation velocity of galaxies to their mass. In contrast to the classical formula, this new relation contains a dependence on the distance of the star to the center of the galaxy. By virtue of this result, it would then be interesting to study whether Kaniadakis statistics can shed any new light on the dark-matter problem.-Kaniadakis entropy has also been applied to the context of quantum information. In particular, in [74], it has been found to exhibit suitable properties to be a candidate for a generalized quantum information theory. Along this direction, a demanding perspective is to explore the possible relevance of Kaniadakis information theory in solving some puzzles arising in quantum gravity scenarios, such as the black hole information paradox and the firewall paradox (see [75] for a recent review on the topic).

The investigation of these and other issues will be performed in separate projects.

## Data Availability

Not applicable.

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
