# Peer review of "Gravity and Cosmology in Kaniadakis Statistics: Current Status and Future Challenges"

_entropy, 2022, doi:10.3390/e24121712_

Round 1
Reviewer 1 Report
The paper is very well written and easily readable. The contents is very timely and relevant. The first part is more like a mini-review on the application of Kaniadakis statistics in cosmology and gravity, and the second part then concentrates on the possible effects of Kaniadakis statistics for BBN, providing quite restrictive bounds via eq. (56) and (64).
I enjoyed reading this excellent paper. However, before acceptance the following points should still be briefly discussed:.
1) The paper concentrates on Kaniadakis entropy, but of course there are many other generalized entropies relevant for physics, see e.g. Contemp. Phys. 50, 495 (2009) for a review on this. Besides Kaniadakis entropy, one has Tsallis entropy, Abe entropy, Landsberg-Vedral entropy, Sharma-Mittal entropy, and many more. All these lead to different deformations of the usual exponential. Why then is Kaniadakis the thing to do for cosmology? At least the author should comment that in principle one could do the same things he does in this paper for other types of entropies as well. So, for example, instead of Kaniadakis-Friedmann eqs. in eq.(29), (30), one could then also study other deformed Friedmann equations. This should at least be mentioned.
2) Further related to point 1): Refs [21] and [34] at least compare Tsallis statistics and Kaniadakis statistics, and the author himself has also a lot of previous papers on Tsallis statistics. The footnote 3 on page 4, which just says that the author does not consider Tsallis statistics here, should perhaps be replaced by something more informative. What would be different if other types of deformations rather than Kaniadakis deformations were used?
3) Minor correction: 6 lines after eq.(12) the author talks about Born-Infield, but I think that should be Born-Infeld.
4) Minor correction: 3 lines after eq.(24) the author cites [41], but I think that should be [42].
Author Response
I am grateful to the Reviewer for his/her positive feedback and comments on the original manuscript.
Concerning Reviewer's comments, please find my point-by-point reply below:
1) + 2) I thank the Reviewer for giving me the possibility to clarify these two related points. I included a blue-highlighted paragraph in the Introduction to explain why I only marginally consider other generalized entropies, in spite of their relevance for many physical applications.
3) I am grateful to the Reviewer for pointing out this typo.
4) I am grateful to the Reviewer for drawing my attention on this point.
I hope that the revised version is now suitable for publication.
Reviewer 2 Report
The research explores recent advances and future challenges of Gravity and Cosmology in Kaniadakis statistics. This is an excellent review, and I recommend it for publication.
Author Response
I am grateful to the Reviewer for his/her positive feedback on the manuscript.
Reviewer 3 Report
Referee Report
Title: Gravity and Cosmology in Kaniadakis statistics: current status and future challenges.
In this paper the authors have analyzed the theoretical and calculated methods recent advances and future challenges of Gravity and Cosmology in Kaniadakis statistics. More specifically, the first part of the work contains a review of κ-entropy implications on Holographic Dark Energy, Entropic Gravity, Black hole thermodynamics and Loop Quantum Gravity, among others. In the second part we focus on the study of Big Bang Nucleosynthesis in Kaniadakis Cosmology. According to the Kaniadakis statistics the relativistic particle systems have explain a non-exponential distribution function with power tails originated from Kaniadakis entropy. The author has analyzed in detail the studies on the Karakadas entropy and its applications to Gravity and Cosmology in Kaniadakis statistical theory, generalized κ entropy on open stellar clusters, Jeans instability and gravitational collapse, Holographic Dark Energy, Entropic Gravity formalism, Black hole thermodynamics and Loop Quantum Gravity.
As a conclusion, I recommend this work for publication as review article.

Author Response

(The authors gave the same response as above.)
